# Factors influencing primary care access, utilisation and quality of management for patients living with hypertension in West Africa: a scoping review protocol

Kezia Naa Amerley Akosua Amarteyfio ,[1] Eugene Paa Kofi Bondzie,[1] Veronika Reichenberger,[2] Irene Akua Agyepong ,[1] Evelyn K Ansah,[1] Aissa Diarra,[3] Tolib Mirzoev ,[2] Pablo Perel,[2] Maurice Yaogo,[4] Edward Antwi[1]

¹Ghana College of Physicians and Surgeons, Accra, Ghana
²London School of Hygiene & Tropical Medicine Centre of Global Change and Health, London, UK
³LASDEL, Niamey, Niger
⁴Universite Catholique de L'Afrique de L'ouest (UCAO) UBB, Bobodialassou, Burkina Faso

**Correspondence to**
Dr Kezia Naa Amerley Akosua Amarteyfio; esikezia@gmail.com

## ABSTRACT

**Introduction** Hypertension, one of the most prevalent non-communicable diseases in West Africa, can be well managed with good primary care. This scoping review will explore what is documented in the literature about factors that influence primary care access, utilisation and quality of management for patients living with hypertension in West Africa.

**Methods and analysis** The scoping review will employ the approach described by Arksey and O'Malley (2005) . The approach has five stages: (1) formulating the research questions, (2) identifying relevant studies, (3) selecting eligible studies, (4) charting the data and (5) collating, summarising and reporting the results. This review will employ the Preferred Reporting Items for Systematic review and Meta-Analysis extension for scoping reviews to report the results. PubMed, Embase, Scopus, Cairn Info and Google Scholar will be searched for publications from 1 January 2000 to 31 December 2023. Studies reported in English, French or Portuguese will be considered for inclusion. Research articles, systematic reviews, observational studies and reports that include information on the relevant factors that influence primary care management of hypertension in West Africa will be eligible for inclusion. Study participants should be adults (aged 18 years or older). Clinical case series/case reports, short communications, books, grey literature and conference proceedings will be excluded. Papers on gestational hypertension and pre-eclampsia will be excluded.

**Ethics and dissemination** This review does not require ethics approval. Our dissemination strategy includes peer-reviewed publications, policy briefs, presentations at conferences, dissemination to stakeholders and intervention co-production forums.

## STRENGTHS AND LIMITATIONS OF THIS STUDY

⇒ This scoping review will be useful for identifying areas of needed intervention and research in the primary care management of hypertension in West Africa.
⇒ This study will also help identify strengths, weaknesses and areas of challenges in the access, utilisation and management of hypertension in primary care facilities.
⇒ There is a possibility of bias because the review will only include studies in the official languages used in West Africa (English, French, and Portuguese).

## INTRODUCTION
### Background

Cardiovascular disease accounts for 17.9 million deaths every year,[1] most of which are in low- and middle-income countries (LMICs). In 2016, for example, 78% of all non-communicable disease (NCD) deaths, and 85% of premature adult NCD deaths, were in LMICs.[2] An estimated 74.7 million individuals in sub-Saharan Africa are hypertensive and this number is projected to increase to 125.5 million by 2025.[3]

Hypertension, also known as raised blood pressure, is diagnosed when blood pressure is measured on two different days and the systolic blood pressure readings on both days is greater or equal to 140 mm Hg and/or the diastolic blood pressure readings on both days is greater or equal to 90 mm Hg.[4] Hypertension is a precursor for various diseases such as ischaemic and haemorrhagic stroke, chronic kidney disease and coronary artery disease.[5] In 2015, 27% of the population in the African region had high blood pressure.[2] Hypertension is common in West Africa.[6] A study on hypertension prevalence, awareness, treatment and control done in seven communities in East and West Africa comprising 3547 participants, showed that a quarter of participants had hypertension, about 40% were unaware, half of those aware were treated and half of those treated had controlled blood pressure.[7] Another study on the rural–urban difference in the prevalence of hypertension in West Africa showed prevalence of

hypertension was 32.6% in men, and 30.0% in women with a lower odd of hypertension in rural compared with urban dwellers.[8]

Primary care is defined by the WHO as 'a model of care that supports first-contact, accessible, continuous, comprehensive and coordinated person-focused care'.[9] It is the most basic form of care available to patients at the base of the healthcare pyramid. It should be accessible to all citizens and be able to provide primary and secondary prevention for common diseases in the community. Primary care is very important because it has been shown to meet the needs of the majority of people throughout their lives.[9] With the increasing emergence of NCDs as a health problem, there is the need to strengthen primary care in their management in the West African subregion. This scoping review explores access, utilisation and quality of primary care for hypertension management to contribute evidence for strengthening primary care for one of the most common NCDs in West Africa.

Healthcare access is 'concerned with helping people to command appropriate health care resources in order to preserve or improve their health'. Access requires adequate service quality and availability or supply, as well as removal of financial, organisational, social and cultural barriers that can limit service use[10] even when it is in adequate supply.[11] Access to healthcare for hypertension in West Africa is often quite low for multiple reasons such as cost, lack of knowledge, essential equipment, tools and supplies as well as medicines.[12 13] People use healthcare for many reasons including preventing and curing health problems, promoting maintenance of health and well-being or obtaining information about their health status and prognosis. Access and utilisation or use of healthcare go hand in hand, as good access to healthcare improves utilisation. Quality of care according to the WHO is the degree to which health services for individuals and populations increase the likelihood of desired health outcomes.[14] Quality primary healthcare should be timely, equitable, integrated and efficient. These three components—access, utilisation and quality of care have intersecting features that come together to ensure that patients receive the care they need. Good access to and utilisation of adequate quality primary care for people living with hypertension can potentially reduce complications and avoidable hospitalisation and mortality.[15 16] Weak and under-resourced health systems and the marginalisation of NCD care in LMICs make it harder to ensure prevention, early diagnosis, access to and utilisation of adequate quality primary care and treatment of patients with hypertension.

The Economic Community of West African States (ECOWAS) subregion is made up of 15 LMICs who share similar social, economic and political environments. The countries in the ECOWAS are Benin, Burkina Faso, Cabo Verde, Cote d'Ivoire, The Gambia, Ghana, Guinea, Guinea-Bissau, Liberia, Mali, Niger, Nigeria, Senegal, Sierra Leone and Togo.[17] A study of 192 441 patients with hypertension in 44 LMICs, including countries in West Africa, to determine the cascade of hypertension care—and its variation between countries and population groups showed that only 18%–29.9% of participants received treatment for hypertension, with 5%–10.3% achieving control of their hypertension.[18] Countries in the Caribbean and Latin America did better than countries in SSA. Hypertension control encompasses not just medications but also preventive practices like lifestyle and dietary changes.[1] To achieve SDG goal 3.4, that is, to reduce by one-third premature mortality from NCDs through prevention and treatment, there is the need to explore the barriers at the primary care level in West Africa.

Contextual factors are a range of factors that make up the wider setting within which the individual and the health system exist. They include personal, social, cultural, economic and political factors that have differing impacts across population groups.[19] In this study, we analyse context using the four categories in Leichter's framework.[20] These are Situational factors such as wars, famine, epidemics, political instability; Structural factors such as economic structure, political system, technological change, degree of urbanisation, structure of the labour markets and demographic structure; Cultural factors such as the level of literacy, and values on issues such as religion, gender, participation and corruption; and Environmental or international/exogenous factors external to the political system such as the role of transnational companies and international agreements and events. These contextual factors influence access to care and individual's decision to seek and maintain treatment for hypertension and are therefore relevant in exploring the barriers to hypertension care at primary care level in West Africa.

Individual, community, primary healthcare facility and health system factors also influence access, quality and utilisation. Some individual/patient factors like medication non-adherence, poor lifestyle and dietary choices, lack of awareness and funds to purchase medications leads to poor management of high blood pressure.[21–23] Health system challenges such as poor referral systems, inadequate tools and equipment, insufficient funding and lack of sufficient health personnel poses challenges in the access to care and quality of management of hypertension at primary care facilities.[22 24 25]

## Rationale

There is an urgent need to explore the available data on the barriers and enablers to the control of hypertension in West Africa especially as part of primary care. Although a similar scoping review was done on the reasons for poor blood pressure control in eastern SSA focusing on patient, professional, primary healthcare system and public health policy challenges,[26] there is no similar study for Western SSA. Also, the study in eastern SSA did not explore contextual factors that influence access, utilisation and quality of primary care for hypertension. For the patients as well as health practitioners, such a review in

West Africa will serve to inform understanding of the challenges in the management of hypertension in primary care facilities. The findings will also be key in informing co-production of interventions as well as identifying areas of needed research in the primary care management of hypertension in West Africa.

## Objectives

This review therefore aims to synthesise what is documented in the literature about the factors that hinder and/or enable primary care for hypertension at individual, contextual, health facility and health system levels in West Africa and how and why they work. This includes access, utilisation as well as quality of available services. The specific objectives of this study are to:

1. Describe the factors documented in the literature that hinder and/or enable management of hypertension at primary care level in West Africa (including access, utilisation and quality of care); and how and why they work.

   1.1. Describe contextual factors (national and global, socio-cultural, socio-economic, political) and analyse how and why they work.

   1.2. Describe individual/patient factors and analyse how and why they work.

   1.3. Describe health facility and health system factors and analyse how and why they work.

2. To identify relevant gaps in the existing literature and areas of research interest, in the primary care of hypertension in West Africa.

## Conceptual framework

A conceptual framework of factors that influence the primary care access, utilisation and quality of care for people living with hypertension in West Africa has been developed to help structure our review as illustrated in figure 1. This framework was adapted from a study on facilitators and barriers to NCD prevention in Pakistan[27] and modified with information from additional studies.[20 28 29]

## METHODS AND ANALYSIS

### Study design

A scoping review methodology as proposed by Arksey and O'Malley[30] will be used due to the broad nature of the research question and the uncertainty of the nature and quality of the evidence available to answer the questions. The approach consists of five stages: (1) formulating the research questions, (2) identifying relevant studies, (3) selecting eligible studies, (4) charting the data and (5) collating, summarising and reporting the results. The scoping review also draws inspiration from methodology developed by the Joanna Briggs Institute.[31] Scoping reviews are useful in presenting a broad overview of the

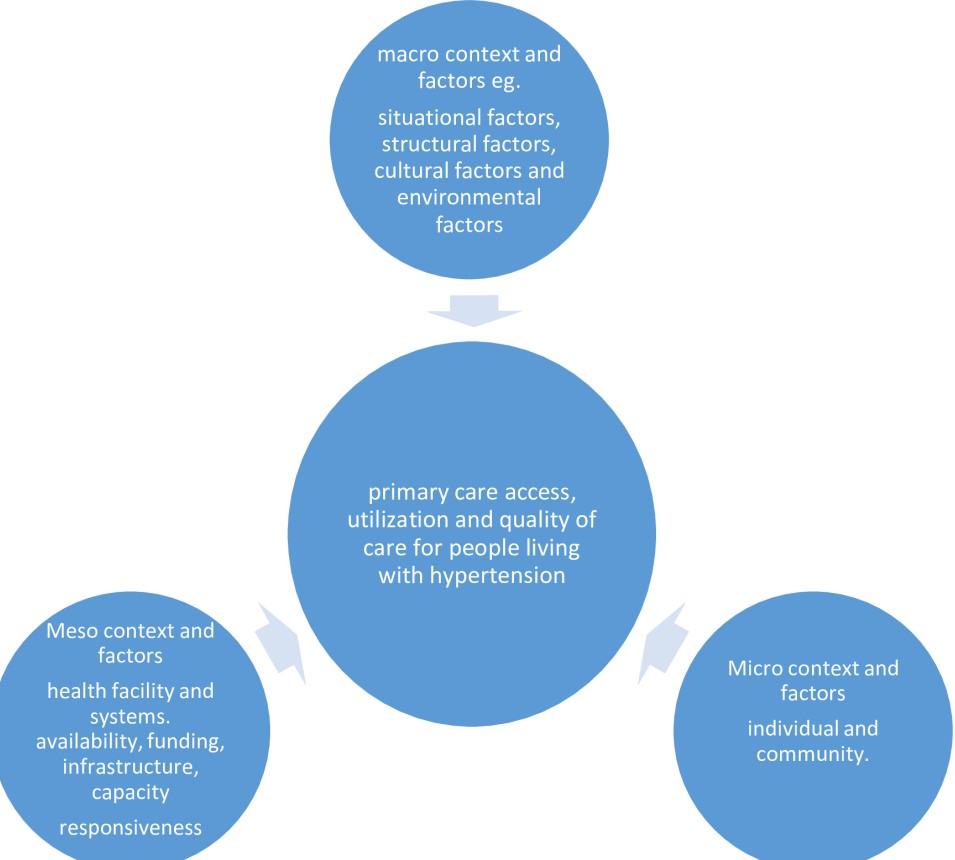

**Figure 1** Conceptual framework on the factors that influence primary care access, utilisation and quality for people living with hypertension.

data available on a topic and allows one to identify gaps in knowledge and research.[32] They also help to determine if a systematic review would be relevant in the area of research. The review will apply the Preferred Reporting Items for Systematic review and Meta-Analysis (PRISMA) extension for scoping reviews to present the results.[33]

The research question was obtained through discussion with key health system stakeholders as well as researchers in the STOP NCD project, which started implementation in October 2022. The protocol was not registered on PROSPERO as PROSPERO currently does not allow for registration of scoping review protocols.

### Identifying the research question
We used an iterative process, as recommended by Arksey and O'Malley, to develop the research question through consultations with the research team and key health system stakeholders. The overall main research question is defined as:

What are the factors (at the individual, community, health system and context level) that influence *access, utilisation and primary care management of people living with hypertension in West Africa* and how and why do they work?

### Identifying relevant studies
Searches will be conducted using PubMed, Embase, Scopus, Cairn Info (primarily for French papers) and Google Scholar databases for the period of 1 January 2000 to 31 December 2023 with the help of a librarian. This time frame was selected because there has been a surge in the burden of NCDs in SSA over the past two decades, driven by increasing incidence of cardiovascular risk factors such as unhealthy diets, reduced physical activity, hypertension, obesity, diabetes, dyslipidaemia and air pollution.[34] The key concepts from our research question and alternative search terms (see online supplemental appendix 1) will be combined using the Boolean terms AND, NOT and OR. Excel power query will be used to generate all combinations of the key search terms to perform an exhaustive search. The search strategies we are using for the various databases is shown in online supplemental appendix 2. The search strategy will be adapted as appropriate for each of the search databases. Only published literature will be included in the review. Other literature may be found using the software 'research rabbit' by inputting relevant papers. Some literature may also be found by searching the reference list and citations of key articles used in the review. Relevant websites such as the WHO website will be accessed for

relevant literature and included in the study. The search results will be exported into Rayyan for removal of duplicates and abstracts screening.

### Study selection
The studies found will be screened against the prespecified eligibility criteria and study selection will be done by two independent researcher pairs. The studies will first be examined with their titles and abstract, and studies that are obviously irrelevant will be removed. Relevant studies will be retrieved and examined fully to determine eligibility. Any disagreements will be resolved through discussion and where necessary, involvement of a third independent researcher. There will be an initial pilot selection of a few papers to test the search eligibility criteria prior to the start of the main selection. This will be done by two independent reviewers to ensure reliability. The final results of the search will be reported in full in the final report and presented in a PRISMA flow diagram[35] (online supplemental appendix 3). The reason for excluding any studies will be documented and included in our final report. Additionally, a description of the PCCS framework, which stands for population, concept, context and study elements, is outlined (table 1) to guide the screening and identification of relevant studies.

### Inclusion criteria
Relevant papers written between 1 January 2000 and 31 December 2023 with participants over the age of 18 years will be included.

Articles, systematic reviews, observational studies and reports that include information on the relevant factors that influence primary care management of hypertension in West Africa will be included in the paper.

Papers written in English, French or Portuguese (Guinea-Bissau and Cape Verde) will be included.

### Exclusion criteria
Clinical case series/case reports, short communications, books, grey literature and conference proceedings will be excluded.

Papers on gestational hypertension and pre-eclampsia will be excluded.

### Data charting
The data to be charted will be entered onto a 'data charting form' using the database programme Excel. The data charting form will include the name of the author, study type, year of publication, study design, study

| Table 1 | Population, concept, context and study elements | | |
|---|---|---|---|
| **Population** | **Concept** | **Context** | **Study** |
| Adults 18 years or older with systemic hypertension in West Africa. | Hindering and enabling factors that influence first level of care for hypertension. | ▶ Macro (national and global level factors). <br> ▶ Meso (health system factors including primary care facilities. <br> ▶ Micro (community factors). | Qualitative, quantitative, meta-analysis, social science case studies and case series. |

population, sample size, study objectives, type of evidence source (ie, primary study, review, website) key findings, comments and limitations. An illustration is shown in online supplemental appendix 4. Two independent reviewer pairs will test the data charting form on a few studies to ensure validity. Any disagreement will be settled by consensus or through the involvement of a third independent reviewer.

## Collating, summarising and reporting the results

The available evidence on the factors that influence primary care for hypertensive patients will be described and summarised under themes with the following headings: individual, contextual, health systems/health facility and gaps in the research. Individual factors include lifestyle, dietary habits and risk factors that impede or encourage management of hypertension. These factors include the patient's adherence to medications and recommended lifestyle changes and barriers to achieving hypertension control. Contextual factors will be explored at macro (national and global levels), meso (subnational including health system and facility levels) and micro (community and household levels). At the macro level this is summarised as the socio-cultural, socio-economic and political environment that hinder/enable primary care management of hypertension using Leichter's framework. At the meso level, the health system and health facility factors are the issues that enable/prevent primary care management for patients living with hypertension. The data collected from the included studies will be tabulated and summarised narratively.

## Patient and public involvement

None.

## Ethics and dissemination

This review does not require ethics approval. Our dissemination strategy includes peer-reviewed publication, policy brief, presentation at conferences, dissemination to relevant stakeholders and intervention co-production forums.

**Acknowledgements** We thank colleagues on the Stop-NCD context analysis team for their comments and feedback on the earlier drafts of this protocol.

**Contributors** KNAAA drafted the research question, study design and methods and final editing, drafting and submission of the protocol. EPKB, VR, IAA, EKA, AD, TM, PP, MY and EA aided in developing the research question and study methods, contributed meaningfully to the drafting and editing and approved the final manuscript.

**Funding** This work received funding from NIHR (National Institute for Health Research) global Health Research centers project: Research and Institutional capacity Strengthening in NCDs. Grant number NIHR203246 – Global Health Research Centre on Strengthening of Capacity for NCD control in West Africa (Stop-NCD).

**Competing interests** None declared.

**Patient and public involvement** Patients and/or the public were not involved in the design, or conduct, or reporting, or dissemination plans of this research.

**Patient consent for publication** Not applicable.

**Provenance and peer review** Not commissioned; externally peer reviewed.

**ORCID iDs**
Kezia Naa Amerley Akosua Amarteyfio http://orcid.org/0009-0005-0383-7927
Irene Akua Agyepong http://orcid.org/0000-0002-0193-5882
Tolib Mirzoev http://orcid.org/0000-0003-2959-9187

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
