## [Reviewer comments · BMJ Open]

ARTICLE DETAILS

TITLE (PROVISIONAL)	Factors influencing primary care access, utilization, and quality of management for patients living with hypertension in West Africa: a scoping review protocol
AUTHORS	Amarteyfio, Kezia; Bondzie, Eugene; Reichenberger, Veronika; Agyepong, Irene Editorial Board Member; Ansah, Evelyn; Diarra, Aissa; Mirzoev, Tolib; Perel, Pablo; Yaogo, Maurice; ANTWI, EDWARD

VERSION 1 – REVIEW

REVIEWER	Arici, Mustafa Hacettepe Universitesi
REVIEW RETURNED	28-Sep-2023

GENERAL COMMENTS	This study protocol paper has defined the aim of the study in detail. My comments are as follows: 1) "Background and extent of the problem" should be more focused on the aim of the study, i.e. use of "primary care access, utilization, and quality of management" rather than the general low control rates of hypertension.2) The authors may enrich their protocol paper by providing more explicit data from the region. They may detail the eastern subsaharan Africa study for better explaining their rationale.3) I don't understand the additional limit that the authors had written. They told that papers written in French will be used, but also mentioned it as a limitation. Please clarify it.4) Some abbreviations must be written in full the first time they are used, such as SDG, SSA.
--

REVIEWER	Sicari, Rosa National Research Council, Institute of Clinical Physiology-Pisa
REVIEW RETURNED	27-Nov-2023

GENERAL COMMENTS	The study protocol to perform an extensive review on hypertensions in West Africa is well designed and appropriate. Methodology is sound and search data bases are appropriate to include the major results in the field. Please avoid the use of grey literature.
---

VERSION 1 – AUTHOR RESPONSE

Reviewer: 1. Dr. Mustafa Arici, Hacettepe Universitesi

1) "Background and extent of the problem" should be more focused on the aim of the study, i.e. use of "primary care access, utilization, and quality of management" rather than the general low control rates of hypertension.

Response: We have included more details about Access, utilization and quality of care for hypertension in the background. (page 4 paragraph 2)

2)The authors may enrich their protocol paper by providing more explicit data from the region. They may detail the eastern sub-Saharan Africa study for better explaining their rationale.

Response: We have included more data from the west Africa/ Sub-Saharan region. (introduction, paragraph 2 and 5)

3) I don't understand the additional limit that the authors had written. They told that papers written in French will be used, but also mentioned it as a limitation. Please clarify it.

Response: On re-examining this statement, we have removed the additional limitation as it is not actually a limitation to the study.

4) Some abbreviations must be written in full the first time they are used, such as SDG, SSA.

Response: Done

Reviewer: 2. Dr. Rosa Sicari, National Research Council

Comments to the Author:

The study protocol to perform an extensive review on hypertensions in West Africa is well designed and appropriate. Methodology is sound and search data bases are appropriate to include the major results in the field. Please avoid the use of grey literature.

Response: Thank you for your comments, we decided to exclude grey literature